# GPyTorch: Blackbox Matrix-Matrix Gaussian Process Inference with GPU Acceleration

**Jacob R. Gardner,**[*] **Geoff Pleiss,**[*]
**David Bindel, Kilian Q. Weinberger, Andrew Gordon Wilson**
Cornell University
{jrg365,kqw4,andrew}@cornell.edu,
{geoff,bindel}@cs.cornell.edu

## Abstract

Despite advances in scalable models, the inference tools used for Gaussian processes (GPs) have yet to fully capitalize on developments in computing hardware. We present an efficient and general approach to GP inference based on Blackbox Matrix-Matrix multiplication (BBMM). BBMM inference uses a modified *batched* version of the conjugate gradients algorithm to derive all terms for training and inference in a single call. BBMM reduces the asymptotic complexity of exact GP inference from $\mathcal{O}(n^3)$ to $\mathcal{O}(n^2)$. Adapting this algorithm to scalable approximations and complex GP models simply requires a routine for efficient matrix-matrix multiplication with the kernel and its derivative. In addition, BBMM uses a specialized preconditioner to substantially speed up convergence. In experiments we show that BBMM effectively uses GPU hardware to dramatically accelerate both exact GP inference and scalable approximations. Additionally, we provide *GPyTorch*, a software platform for scalable GP inference via BBMM, built on PyTorch.

## 1 Introduction

The past years have witnessed unprecedented innovation in deep learning. This progress has involved innovations in network designs [18, 20, 24, 26, 30], but it also has benefited vastly from improvements in optimization [6], and excellent software implementations such as PyTorch, MXNet, TensorFlow and Caffe [1, 8, 28, 38]. Broadly speaking, the gains in optimization originate in large part from insights in stochastic gradient optimization [6, 7, 23, 27, 29, 31], effectively trading off unnecessary exactness for speed and in some cases regularization. Moreover, the advantages of modern software frameworks for deep learning include rapid prototyping, easy access to specialty compute hardware (such as GPUs), and blackbox optimization through automatic differentiation.

Similarly, Gaussian process research has undergone significant innovations in recent years [9, 21, 45, 49–51] — in particular to improve scalability to large data sets. However, the tools most commonly used for GP inference do not effectively utilize modern hardware, and new models require significant implementation efforts. Often, in fact, the *model* and the *inference engine* are tightly coupled and consequently many complex models like multi-output GPs and scalable GP approximations require custom inference procedures [5, 22]. This entanglement of model specification and inference procedure impedes rapid prototyping of different model types, and obstructs innovation in the field.

In this paper, we address this gap by introducing a highly efficient framework for Gaussian process inference. Whereas previous inference approaches require the user to provide routines for computing the full GP marginal log likelihood for a sufficiently complex model, our framework only requires access to a blackbox routine that performs matrix-matrix multiplications with the kernel matrix and its derivative. Accordingly, we refer to our method as Blackbox Matrix-Matrix (BBMM) Inference.

---

[*]Equal contribution.

In contrast to the Cholesky decomposition, which is at the heart of many existing inference engines, matrix-matrix multiplications fully utilize GPU acceleration. We will demonstrate that this matrix-matrix approach also significantly eases implementation for a wide class of existing GP models from the literature. In particular, we make the following contributions:

1. Inspired by iterative matrix-vector multiplication (MVM)-based inference methods [9, 13, 43, 50, 51], we provide a modified *batched* version of linear conjugate gradients (mBCG) that provides all computations necessary for both the marginal likelihood and its derivatives. Moreover, mBCG uses large matrix-matrix multiplications that more efficiently utilize modern hardware than both existing Cholesky and MVM based inference strategies. Our approach also circumvents several critical space complexity and numerical stability issues present in existing inference methods. Most notably, BBMM reduces the time complexity of exact GP inference from $\mathcal{O}(n^3)$ to $\mathcal{O}(n^2)$.

2. We introduce a method for *preconditioning* this modified conjugate gradients algorithm based on the pivoted Cholesky decomposition [4, 19]. All required operations with this preconditioner are efficient, and in practice require negligible time. We demonstrate both empirically and theoretically that this preconditioner significantly accelerates inference.

3. We introduce **GPyTorch**, a new software platform using BBMM inference for scalable Gaussian processes, which is built on top of PyTorch: https://gpytorch.ai. On datasets as large as 3000 data points (until we fill GPU memory) we demonstrate that *exact* GPs with BBMM are *up to $20\times$ faster than GPs using Cholesky-based approaches*. Moreover, the popular SKI [50] and SGPR [45] frameworks with BBMM achieve up to $15\times$ and $4\times$ speedups (respectively) on datasets as large as 500,000 data points. Additionally, SKI, SGPR and other scalable approximations are implemented in *less than 50 lines of code*, requiring only an efficient matrix-matrix multiplication routine.

## 2 Related Work

**Conjugate gradients, the Lanczos tridiagonalization algorithm,** and their relatives are methods from numerical linear algebra for computing linear solves and solving eigenvalue problems *without explicitly computing a matrix*. These techniques have been around for decades, and are covered in popular books and papers [11, 12, 17, 32, 36, 37, 42]. These algorithms belong to a broad class of iterative methods known as *Krylov subspace methods*, which access matrices only through matrix-vector multiplies (MVMs). Historically, these methods have been applied to solving large numerical linear algebra problems, particularly those involving sparse matrices that afford fast MVMs.

Recently, a number of papers have used these MVM methods for parts of GP inference [9, 13, 15, 34, 39, 43, 49, 50]. One key advantage is that MVM approaches can exploit algebraic structure for increased computational efficiencies. Notably, the structured kernel interpolation (SKI) method [50] uses structured kernel matrices with fast MVMs to achieve a remarkable asymptotic complexity. Dong et al. [13] propose MVM methods for computing stochastic estimates of log determinants and their derivatives using a technique based on Lanczos tridiagonalization [16, 46]. We utilize the same log determinant estimator as Dong et al. [13], except we avoid explicitly using the Lanczos tridiagonalization algorithm which has storage and numerical stability issues [17].

**Preconditioning** is an effective tool for accelerating the convergence of conjugate gradients. These techniques are far too numerous to review adequately here; however, Saad [42] contains two chapters discussing a variety of preconditioning techniques. Cutajar et al. [10] explores using preconditioned conjugate gradients for exact GP inference, where they use various sparse GP methods (as well as some classical methods) as preconditioners. However, the methods in Cutajar et al. [10] do not provide general purpose preconditioners. For example, methods like Jacobi preconditioning have no effect when using a stationary kernel [10, 51], and many other preconditioners have $\Omega(n^2)$ complexity, which dominates the complexity of most scalable GP methods.

**The Pivoted Cholesky decomposition** is an efficient algorithm for computing a low-rank decomposition of a positive definite matrix [4, 19], which we use in the context of preconditioning. Harbrecht et al. [19] explores the use of the pivoted Cholesky decomposition as a low rank approximation, although primarily in a scientific computing context. In proving convergence bounds for our preconditioner we explicitly make use of some theoretical results from [19] (see Appendix D). Bach [4] considers using random column sampling as well as the pivoted Cholesky decomposition as a low-rank approximation to kernel matrices. However, Bach [4] treats this decomposition as an

approximate training method, whereas we use the pivoted Cholesky decomposition primarily as a preconditioner, which avoids any loss of accuracy from the low rank approximation as well as the complexity of computing derivatives.

# 3 Background

**Notation.** $X$ will denote a set of $n$ training examples in $d$ dimensions, or equivalently an $n \times d$ matrix where the $i^{\text{th}}$ row (denoted $\mathbf{x}_i$) is the $i^{\text{th}}$ training example. $\mathbf{y}$ denotes the training labels. $k(\mathbf{x}, \mathbf{x}')$ denotes a *kernel function*, and $K_{XX}$ denotes the matrix containing all pairs of kernel entries, i.e. $[K_{XX}]_{ij} = k(\mathbf{x}_i, \mathbf{x}_j)$. $\mathbf{k}_{X\mathbf{x}^*}$ denotes kernel values between training examples and a test point $\mathbf{x}^*$, e.g. $[\mathbf{k}_{X\mathbf{x}^*}]_i = k(\mathbf{x}_i, \mathbf{x}^*)$. A hat denotes an added diagonal: $\widehat{K}_{XX} = K_{XX} + \sigma^2 I$.

**A Gaussian process** (GP) is a kernel method that defines a full distribution over the function being modeled, $f(\mathbf{x}) \sim \mathcal{GP}(\mu(\mathbf{x}), k(\mathbf{x}, \mathbf{x}'))$. Popular kernels include the RBF kernel, $k(\mathbf{x}, \mathbf{x}') = s \exp\left(-(\|\mathbf{x} - \mathbf{x}'\|)/(2\ell^2)\right)$ and the Matérn family of kernels [41].

**Predictions with a Gaussian process.** Predictions with a GP are made utilizing the *predictive posterior distribution*, $p(f(\mathbf{x}^*) \mid X, \mathbf{y})$. Given two test inputs $\mathbf{x}^*$ and $\mathbf{x}^{*\prime}$, the predictive mean for $\mathbf{x}^*$ and the predictive covariance between $\mathbf{x}^*$ and $\mathbf{x}^{*\prime}$ are given by:

$$\mu_{f|\mathcal{D}}(\mathbf{x}^*) = \mu(\mathbf{x}^*) + \mathbf{k}_{X\mathbf{x}^*}^\top \widehat{K}_{XX}^{-1} \mathbf{y}, \qquad k_{f|\mathcal{D}}(\mathbf{x}^*, \mathbf{x}^{*\prime}) = k_{\mathbf{x}^*\mathbf{x}^{*\prime}} - \mathbf{k}_{X\mathbf{x}^*}^\top \widehat{K}_{XX}^{-1} \mathbf{k}_{X\mathbf{x}^{*\prime}}, \quad (1)$$

**Training a Gaussian process.** Gaussian processes depend on a number of *hyperparameters* $\theta$. Hyperparameters may include the likelihood noise, kernel lengthscale, inducing point locations [45], or neural network parameters for deep kernel learning [52]. These parameters are commonly learned by minimization or sampling via the *negative log marginal likelihood*, given (with derivative) by

$$L(\theta \mid X, \mathbf{y}) \propto \log \left| \widehat{K}_{XX} \right| - \mathbf{y}^\top \widehat{K}_{XX}^{-1} \mathbf{y}, \quad \frac{dL}{d\theta} = \mathbf{y}^\top \widehat{K}_{XX}^{-1} \frac{d\widehat{K}_{XX}}{d\theta} \widehat{K}_{XX}^{-1} \mathbf{y} + \text{Tr}\left( \widehat{K}_{XX}^{-1} \frac{d\widehat{K}_{XX}}{d\theta} \right). \quad (2)$$

# 4 Gaussian process inference through blackbox matrix multiplication

The goal of our paper is to replace existing inference strategies with a unified framework that utilizes modern hardware efficiently. We additionally desire that complex GP models can be used in a blackbox manner without additional inference rules. To this end, our method reduces the bulk of GP inference to one of the most efficiently-parallelized computations: *matrix-matrix multiplication*. We call our method Blackbox Matrix-Matrix inference (BBMM) because it only requires a user to specify a matrix multiply routine for the kernel $\widehat{K}_{XX} M$ and its derivative $\frac{d\widehat{K}_{XX}}{d\theta} M$.

**Required operations.** An *inference engine* is a scheme for computing all the equations discussed above: the predictive distribution (1), the loss, and its derivative (2). These equations have three operations in common that dominate its time complexity: 1) the linear solve $\widehat{K}_{XX}^{-1} \mathbf{y}$, 2) the log determinant $\log |\widehat{K}_{XX}|$, and 3) a trace term $\text{Tr}(\widehat{K}_{XX}^{-1} \frac{d\widehat{K}_{XX}}{d\theta})$. In many implementations, these three quantities are computed using the Cholesky decomposition of $\widehat{K}_{XX}$, which is computationally expensive, requiring $\mathcal{O}(n^3)$ operations, and does not effectively utilize parallel hardware.

Recently, there is a growing line of research that computes these operations with iterative routines based on matrix-vector multiplications (MVMs). $\widehat{K}_{XX}^{-1} \mathbf{y}$ can be computed using *conjugate gradients* (CG) [9, 10, 43, 50], and the other two quantities can be computed using calls to the iterative Lanczos tridiagonalization algorithm [13, 46]. MVM-based methods are asymptotically faster and more space efficient than Cholesky based methods [13, 50]. Additionally, these methods are able to exploit algebraic structure in the data for further efficiencies [9, 43, 50]. However, they also have disadvantages. The quantities are computed via several independent calls to the CG and stochastic Lanczos quadrature subroutines, which are inherently sequential and therefore do not fully utilize parallel hardware. Additionally, the Lanczos tridiagonalization algorithm requires $\mathcal{O}(np)$ space for $p$ iterations and suffers from numerical stability issues due to loss of orthogonality [17].

**Modified CG.** Our goal is to capitalize on the advantages of MVM-based methods (space-efficiency, ability to exploit structure, etc.) but with efficient routines that are optimized for modern parallel

compute hardware. For this purpose, our method makes use of a *modified Batched Conjugate Gradients Algorithm* (mBCG) algorithm. Standard conjugate gradients takes as input a vector $\mathbf{y}$ and a routine for computing a matrix vector product $\widehat{K}_{XX}\mathbf{y}$, and, after $p$ iterations, outputs an approximate solve $\mathbf{u}_p \approx \widehat{K}_{XX}^{-1}\mathbf{y}$ (with exact equality when $p = n$). We modify conjugate gradients to (1) perform linear solves with multiple right hand sides simultaneously, and (2) return tridiagonal matrices corresponding to partial Lanczos tridiagonalizations of $\widehat{K}_{XX}$ with respect to each right hand side.[2] Specifically, mBCG takes as input a matrix $[\ \mathbf{y}\ \ \mathbf{z}_1\ \ \cdots\ \ \mathbf{z}_t\ ]$, and outputs:

$$[\ \mathbf{u}_0\ \ \mathbf{u}_1\ \ \cdots\ \ \mathbf{u}_t\ ] = \widehat{K}_{XX}^{-1}[\ \mathbf{y}\ \ \mathbf{z}_1\ \ \cdots\ \ \mathbf{z}_t\ ] \quad \text{and} \quad \tilde{T}_1, ..., \tilde{T}_t \tag{3}$$

where $\tilde{T}_1, \ldots, \tilde{T}_t$ are partial Lanczos tridiagonalizations of $\widehat{K}_{XX}$ with respect to the vectors $\mathbf{z}_1, \ldots, \mathbf{z}_t$, which we describe shortly. In what follows, we show how to use a single call to mBCG to compute the three GP inference terms: $\widehat{K}_{XX}^{-1}\mathbf{y}$, $\text{Tr}(\widehat{K}_{XX}^{-1}\frac{\partial \widehat{K}_{XX}}{\partial \theta})$, and $\log|\widehat{K}_{XX}|$. $\widehat{K}_{XX}^{-1}\mathbf{y}$ is equal to $\mathbf{u}_0$ in (3), directly returned from mBCG. We describe the other two terms below.

**Estimating $\text{Tr}(\widehat{K}_{XX}^{-1}\frac{\partial \widehat{K}_{XX}}{\partial \theta})$** from CG relies on *stochastic trace estimation* [3, 14, 25], which allows us to treat this term as a sum of linear solves. Given i.i.d. random variables $\mathbf{z}_1, \ldots, \mathbf{z}_t$ so that $\mathbb{E}[\mathbf{z}_i] = 0$ and $\mathbb{E}[\mathbf{z}_i\mathbf{z}_i^\top] = I$, (e.g., $\mathbf{z}_i \sim \mathcal{N}(0, I)$) the matrix trace $\text{Tr}(A)$ can be written as $\text{Tr}(A) = \mathbb{E}[\mathbf{z}_i^\top A \mathbf{z}_i]$, such that

$$\text{Tr}\left(\widehat{K}_{XX}^{-1}\frac{d\widehat{K}_{XX}}{d\theta}\right) = \mathbb{E}\left[\mathbf{z}_i^\top \widehat{K}_{XX}^{-1}\frac{d\widehat{K}_{XX}}{d\theta}\mathbf{z}_i\right] \approx \frac{1}{t}\sum_{i=1}^{t}\left(\mathbf{z}_i^\top \widehat{K}_{XX}^{-1}\right)\left(\frac{d\widehat{K}_{XX}}{d\theta}\mathbf{z}_i\right) \tag{4}$$

is an unbiased estimator of the derivative. This computation motivates the $\mathbf{z}_1, \ldots, \mathbf{z}_t$ terms in (3): the mBCG call returns the solves $\widehat{K}_{XX}^{-1}[\mathbf{z}_1 \ldots \mathbf{z}_t]$, which yields $\mathbf{u}_i = \mathbf{z}_i^\top \widehat{K}_{XX}^{-1}$. A single matrix multiply with the derivative $\frac{d\widehat{K}_{XX}}{d\theta}[\mathbf{z}_1 \ldots \mathbf{z}_t]$ yields the remaining terms on the RHS. The full trace can then be estimated by elementwise multiplying these terms together and summing, as in (4).

**Estimating $\log|\widehat{K}_{XX}|$** can be accomplished using the $T_1, ..., T_t$ matrices from mBCG. If $\widehat{K}_{XX} = QTQ^\top$, with $Q$ orthonormal, then because $\widehat{K}_{XX}$ and $T$ have the same eigenvalues:

$$\log|\widehat{K}_{XX}| = \text{Tr}\left(\log T\right) = \mathbb{E}\left[\mathbf{z}_i^\top (\log T)\mathbf{z}_i\right] \approx \sum_{i=1}^{t}\mathbf{z}_i^\top (\log T)\,\mathbf{z}_i \tag{5}$$

where $\log T$ here denotes the matrix logarithm, and the approximation comes from the same stochastic trace estimation technique used for (4). One approach to obtain a decomposition $\widehat{K}_{XX} = QTQ^\top$ is to use the *Lanczos tridiagonalization algorithm*. This algorithm takes the matrix $\widehat{K}_{XX}$ and a probe vector $\mathbf{z}$ and outputs the decomposition $QTQ^\top$ (where $\mathbf{z}$ is the first column of $Q$). However, rather than running the full algorithm, we can instead run $p$ iterations of the algorithm $t$ times, each with a vector $\mathbf{z}_1, ..., \mathbf{z}_t$ to obtain $t$ decompositions $\tilde{Q}_1\tilde{T}_1\tilde{Q}_1^\top, ..., \tilde{Q}_t\tilde{T}_t\tilde{Q}_t^\top$ with $\tilde{Q}_i \in \mathbb{R}^{n \times p}$ and $\tilde{T}_i \in \mathbb{R}^{p \times p}$. We can use these partial decompositions to estimate (5):

$$\mathbb{E}\left[\mathbf{z}_i^\top (\log T)\mathbf{z}_i\right] = \mathbb{E}\left[\mathbf{z}_i^\top \tilde{Q}_i(\log \tilde{T}_i)\tilde{Q}_i^\top \mathbf{z}_i\right] \approx \frac{1}{t}\sum_{i=1}^{t}\mathbf{z}_i^\top \tilde{Q}_i(\log \tilde{T}_i)\tilde{Q}_i^\top \mathbf{z}_i = \frac{1}{t}\sum_{i=1}^{t}e_1^\top (\log \tilde{T}_i)e_1, \tag{6}$$

where $e_1$ is the first row of the identity matrix. Running Lanczos with a starting vector $\mathbf{z}_i$ ensures that all columns of $\tilde{Q}_i$ are orthogonal to $\mathbf{z}_i$ except the first, so $\tilde{Q}_i\mathbf{z}_i = e_1$ [13, 16, 46].

In mBCG, we adapt a technique from Saad [42] which allows us to compute $\tilde{T}_1, \ldots, \tilde{T}_t$ corresponding to the input vectors $\mathbf{z}_1, \ldots, \mathbf{z}_t$ to mBCG from the coefficients of CG in $\mathcal{O}(1)$ additional work per iteration. This approach allows us to compute a log determinant estimate identical to (6) *without running the Lanczos algorithm*. Thus we avoid the extra computation, storage, and numerical instability associated with Lanczos iterations. We describe the details of this adaptation in Appendix A.

**Runtime and space.** As shown above, we are able to approximate all inference terms from a single call to mBCG. These approximations improve with the number of mBCG iterations. Each iteration

requires one matrix-matrix multiply with $\widehat{K}_{XX}$, and the subsequent work to derive these inference terms takes negligible additional time (Appendix B). Therefore, $p$ iterations of mBCG requires $\mathcal{O}(nt)$ space (see Appendix B) and $\mathcal{O}(p\,\Xi(\widehat{K}_{XX}))$ time, where $\Xi(\widehat{K}_{XX})$ is the time to multiply $\widehat{K}_{XX}$ by a $n \times t$ matrix. This multiplication takes $\mathcal{O}(n^2 t)$ time with a standard matrix. It is worth noting that this is a lower asymptotic complexity that standard Cholesky-based inference, which is $\mathcal{O}(n^3)$. Therefore, BBMM offers a computational speedup for exact GP inference. As we will show in Section 5, this time complexity can be further reduced with structured data or sparse GP approximations.

## 4.1 Preconditioning

While each iteration of mBCG performs large parallel matrix-matrix operations that utilize hardware efficiently, the iterations themselves are sequential. A natural goal for better utilizing hardware is to trade off fewer sequential steps for slightly more effort per step. We accomplish this goal using *preconditioning* [12, 17, 42, 47], which introduces a matrix $P$ to solve the related linear system

$$P^{-1}\widehat{K}_{XX}\mathbf{u} = P^{-1}\mathbf{y}$$

instead of $\widehat{K}_{XX}^{-1}\mathbf{y}$. Both systems are guaranteed to have the same solution, but the preconditioned system's convergence depends on the conditioning of $P^{-1}\widehat{K}_{XX}$ rather than that of $\widehat{K}_{XX}$.

We observe two requirements of a preconditioner to be used in general for GP inference. First, in order to ensure that preconditioning operations do not dominate running time when using scalable GP methods, the preconditioner should afford roughly linear time solves and space. Second, we should be able to efficiently compute the log determinant of the preconditioner matrix, $\log|P|$. This is because the mBCG algorithm applied to the preconditioned system estimates $\log|P^{-1}\widehat{K}_{XX}|$ rather than $\log|\widehat{K}_{XX}|$. We must therefore compute $\log|\widehat{K}_{XX}| = \log|P^{-1}\widehat{K}_{XX}| + \log|P|$.

**The Pivoted Cholesky Decomposition.** For one possible preconditioner, we turn to the *pivoted Cholesky* decomposition. The pivoted Cholesky algorithm allows us to compute a low-rank approximation of a positive definite matrix, $K_{XX} \approx L_k L_k^\top$ [19]. We precondition mBCG with $(L_k L_k^\top + \sigma^2 I)^{-1}$, where $\sigma^2$ is the Gaussian likelihood's noise term. Intuitively, if $P_k = L_k L_k^\top$ is a good approximation of $K_{XX}$, then $(P_k + \sigma^2 I)^{-1}\widehat{K}_{XX} \approx I$.

While we review the pivoted Cholesky algorithm fully in Appendix C, we would like to emphasize three key properties. First, it can be computed in $\mathcal{O}(\rho(K_{XX})k^2)$ time, where $\rho(K_{XX})$ is the time to access a row (nominally this is $\mathcal{O}(n)$). Second, linear solves with $\widehat{P} = L_k L_k^\top + \sigma^2 I$ can be performed in $\mathcal{O}(nk^2)$ time. Finally, the log determinant of $\widehat{P}$ can be computed in $\mathcal{O}(nk^2)$ time. In Figure 6 we empirically show that this preconditioner dramatically accelerates CG convergence. Further, in Appendix D, we prove the following lemma and theorem for univariate RBF kernels:

**Lemma 1.** *Let $K_{XX} \in \mathbb{R}^{n\times n}$ be a univariate RBF kernel matrix. Let $L_k L_k^\top$ be the rank $k$ pivoted Cholesky decomposition of $K_{XX}$, and let $\widehat{P}_k = L_k L_k^\top + \sigma^2 I$. Then there exists a constant $b > 0$ so that the condition number $\kappa(\widehat{P}^{-1}\widehat{K}_{XX})$ satisfies the following inequality:*

$$\kappa\left(\widehat{P}_k^{-1}\widehat{K}_{XX}\right) \triangleq \left\|\widehat{P}_k^{-1}\widehat{K}_{XX}\right\|_2 \left\|\widehat{K}_{XX}^{-1}\widehat{P}_k\right\|_2 \le (1 + \mathcal{O}(n\exp(-bk)))^2. \tag{7}$$

**Theorem 1** (Convergence of pivoted Cholesky-preconditioned CG). *Let $K_{XX} \in \mathbb{R}^{n\times n}$ be a $n \times n$ univariate RBF kernel, and let $L_k L_k^\top$ be its rank $k$ pivoted Cholesky decomposition. Assume we are using preconditioned CG to solve the system $\widehat{K}_{XX}^{-1}\mathbf{y} = (K_{XX} + \sigma^2 I)^{-1}\mathbf{y}$ with preconditioner $\widehat{P} = (L_k L_k^\top + \sigma^2 I)$. Let $\mathbf{u}_p$ be the $p^{th}$ solution of CG, and let $\mathbf{u}^* = \widehat{K}_{XX}^{-1}\mathbf{y}$ be the exact solution. Then there exists some $b > 0$ such that:*

$$\|\mathbf{u}^* - \mathbf{u}_p\|_{\widehat{K}_{XX}} \le 2\left(1/(1 + \mathcal{O}(\exp(kb)/n))\right)^p \|\mathbf{u}^* - \mathbf{u}_0\|_{\widehat{K}_{XX}}. \tag{8}$$

Theorem 1 implies that we should expect the convergence of conjugate gradients to improve *exponentially* with the rank of the pivoted Cholesky decomposition used for RBF kernels. In our experiments we observe significantly improved convergence for other kernels as well (Figure 6). Furthermore, we can leverage Lemma 1 and existing theory from [46] to argue that preconditioning improves our log determinant estimate. In particular, we restate Theorem 4.1 of Ubaru et al. [46] here:

**Theorem 2** (Theorem 4.1 of Ubaru et al. [46]). *Let $K_{XX} \in \mathbb{R}^{n \times n}$, and let $L_k L_k^\top$ be its rank $k$ pivoted Cholesky decomposition. Suppose we run $p \geq \frac{1}{4} \sqrt{\kappa \left( \widehat{P}_k^{-1} \widehat{K}_{XX} \right)} \log \frac{D}{\epsilon}$ iterations of mBCG, where $D$ is a term involving this same condition number that vanishes as $k \to n$ (see [46]), and we use $t \geq \frac{24}{\epsilon^2} \log(2/\delta)$ vectors $\mathbf{z}_i$ for the solves. Let $\Gamma$ be the log determinant estimate from (6). Then:*

$$Pr \left[ |\log |\widehat{P}^{-1} \widehat{K}_{XX}| - \Gamma| \leq \epsilon |\log |\widehat{P}^{-1} \widehat{K}_{XX}|| \right] \geq 1 - \delta. \tag{9}$$

Because Lemma 1 states that the condition number $\kappa \left( \widehat{P}_k^{-1} \widehat{K}_{XX} \right)$ decays exponentially with the rank of $L_k$, Theorem 2 implies that we should expect that the number of CG iterations required to accurately estimate $\log |\widehat{P}^{-1} \widehat{K}_{XX}|$ decreases quickly as $k$ increases. In addition, in the limit as $k \to n$ we have that $\log |\widehat{K}_{XX}| = \log |\widehat{P}|$. This is because $\log |\widehat{P}^{-1} \widehat{K}_{XX}| \to 0$ (since $\widehat{P}^{-1} \widehat{K}_{XX}$ converges to $I$) and we have that $\log |\widehat{K}_{XX}| = \log |\widehat{P}^{-1} \widehat{K}_{XX}| + \log |\widehat{P}|$. Since our calculation of $\log |\widehat{P}|$ is exact, our final estimate of $\log |\widehat{K}_{XX}|$ becomes more exact as $k$ increases. In future work we hope to derive a more general result that covers multivariate settings and other kernels.

## 5  Programmability with BBMM

We have discussed how the BBMM framework is more hardware efficient than existing inference engines, and avoids numerical instabilities with Lanczos. Another key advantage of BBMM is that it can easily be adapted to complex GP models or structured GP approximations.

Indeed BBMM is *blackbox* by nature, only requiring a routine to perform matrix-multiplications with the kernel matrix and its derivative. Here we provide examples of how existing GP models and scalable approximations can be easily implemented in this framework. The matrix-multiplication routines for the models require at most *50 lines of Python code*. All our software, including the following GP implementations with BBMM, are available through our GPyTorch library: https://gpytorch.ai.

**Bayesian linear regression** can be viewed as GP regression with the special kernel matrix $\widehat{K}_{XX} = XX^\top + \sigma^2 I$. A matrix multiply with this kernel against an $n \times t$ matrix $V$, $(XX^\top + \sigma^2 I)V$ requires $\mathcal{O}(tnd)$ time. Therefore, BBMM requires $\mathcal{O}(ptnd)$ time, and is exact in $\mathcal{O}(tnd^2)$ time. This running time complexity matches existing efficient algorithms for Bayesian linear regression, *with no additional derivation*. Multi-task Gaussian processes [5] can be adapted in the same fashion [15].

**Sparse Gaussian Process Regression (SGPR)** [45] and many other sparse GP techniques [21, 40, 44] use the subset of regressors (SoR) approximation for the kernel: $\widehat{K}_{XX} \approx (K_{XU} K_{UU}^{-1} K_{UX} + \sigma^2 I)$. Performing a matrix-matrix multiply with this matrix requires $\mathcal{O}(tnm + tm^3)$ time by distributing the vector multiply and grouping terms correctly. This computation is *asymptotically faster* than the $\mathcal{O}(nm^2 + m^3)$ time required by Cholesky based inference. Augmenting the SoR approximation with a diagonal correction, e.g. as in FITC [44], is similarly straightforward.

**Structured Kernel Interpolation (SKI)** [50], also known as KISS-GP, is an inducing point method designed to provide fast matrix vector multiplies (MVMs) for use with Krylov subspace methods. SKI is thus a natural candidate for BBMM and can benefit greatly from hardware acceleration. SKI is a generalization of SoR, which specifies $K_{XU} \approx W K_{UU}$, where $W$ is a sparse matrix. For example $W$ can correspond to the coefficients of sparse local cubic convolution interpolation. The SKI approximation applied to the training covariance matrix gives us $\widehat{K}_{XX} \approx (W K_{UU} W^\top + \sigma^2 I)$. Assuming no structure in $K_{UU}$ a matrix multiply requires $\mathcal{O}(tn + tm^2)$ time. In KISS-GP [50, 51], the matrix $K_{UU}$ is also chosen to have algebraic structure, such as Kronecker or Toeplitz structure, which further accelerates MVMs. For example, MVMs with a Toeplitz $K_{UU}$ only require $\mathcal{O}(m \log m)$ time. Thus KISS-GP provides $\mathcal{O}(tn + tm \log m)$ matrix-matrix multiplies [50].

**Compositions of kernels** can often be handled automatically. For example, given a BBMM routine for $K_1, K_2, K_3$, we can automatically perform $(K_1 K_2 + K_3)M = K_1(K_2 M) + K_3 M$. SGPR and KISS-GP are implemented in this fashion. Given some pre-defined basic compositionality strategies, the kernel matrix multiplication $KM$ in SGPR reduces to defining how to perform $K_{UU}^{-1} M$, and

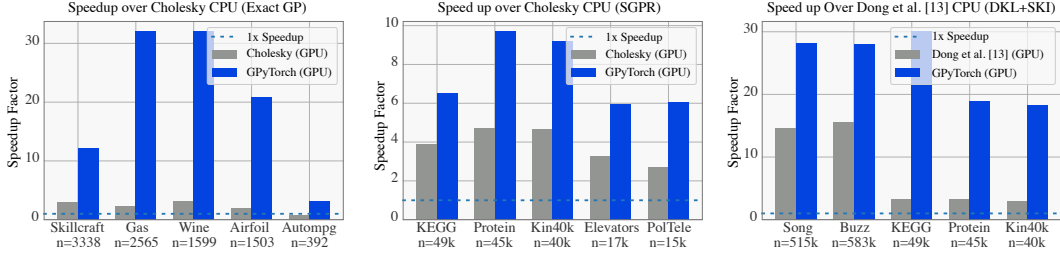

Figure 1: Speedup of GPU-accelerated inference engines. BBMM is in blue, and competing GPU methods are in gray. **Left:** Exact GPs. **Middle:** SGPR [21, 45] – speedup over CPU Cholesky-based inference engines. **Right:** SKI+DKL [50, 52] – speedup over CPU inference of Dong et al. [13].

similarly for KISS-GP it reduces to performing multiplication with a Toeplitz matrix $K_{UU}M$. For product kernels one can follow Gardner et al. [15].

## 6 Results

We evaluate the BBMM framework, demonstrating: (1) the BBMM inference engine provides a substantial speed benefit over Cholesky based inference and standard MVM-based CG inference, especially for GPU computing; (2) BBMM achieves comparable or better final test error compared to Cholesky inference, even with no kernel approximations; and (3) preconditioning provides a substantial improvement in the efficiency of our approach.

**Baseline methods.** We test BBMM on three types of GPs: 1. **Exact** GP models, 2. **SGPR** inducing point models [21, 45], and 3. **SKI** models with Toeplitz $K_{UU}$ and deep kernels [50, 52]. For Exact and SGPR, we compare BBMM against Cholesky-based inference engines implemented in GPFlow [33]. GPFlow is presently the fastest implementation of these models with a Cholesky inference engine. Since SKI is not intended for Cholesky inference, we compare BBMM to the inference procedure of Dong et al. [13], implemented in our GPyTorch package. This procedure differers from BBMM in that it computes $\widehat{K}_{XX}^{-1}\mathbf{y}$ without a preconditioner and computes $\log|\widehat{K}_{XX}|$ and its derivative with the Lanczos algorithm.

**Datasets.** We test Exact models on five datasets from the UCI dataset repository [2] with up to 3500 training examples (the largest possible before all implementations exhausted GPU memory): Skillcraft, Gas, Airfoil, Autompg, and Wine. We test SGPR on larger datasets ($n$ up to 50000): KEGG, Protein, Elevators, Kin40k, and PoleTele. For SKI we test five of the largest UCI datasets ($n$ up to 515000): Song, Buzz, Protein, Kin40k, and KEGG.

**Experiment details.** All methods use the same optimizer (Adam) with identical hyperparameters. In BBMM experiments we use rank $k = 5$ pivoted Cholesky preconditioners unless otherwise stated. We use a maximum of $p = 20$ iterations of CG for each solve, and we use $t = 10$ probe vectors filled with Rademacher random variables to estimate the log determinant and trace terms. SGPR models use 300 inducing points. SKI models use 10,000 inducing points and the deep kernels described in [52]. The BBMM inference engine is implemented in our GPyTorch package. All speed experiments are run on an Intel Xeon E5-2650 CPU and an NVIDIA Titan Xp GPU.

**Speed comparison.** Figure 1 shows the speedup obtained by GPU-accelerated BBMM over the leading CPU-based inference engines (Cholesky for Exact/SGPR, Dong et al. [13] for SKI). As would be expected, GPU-accelerated BBMM is faster than CPU-based inference. On Exact and SKI, BBMM is up to *32 times faster* than CPU inference, and up to 10 times faster on SGPR. The largest speedups occur on the largest datasets, since smaller datasets experience larger GPU overhead. Notably, BBMM achieves a much larger speedup than GPU ac-

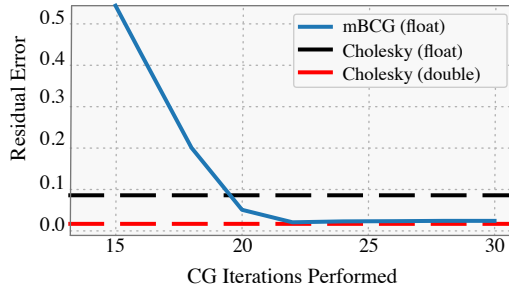

Figure 2: Solve error for mBCG and Cholesky.

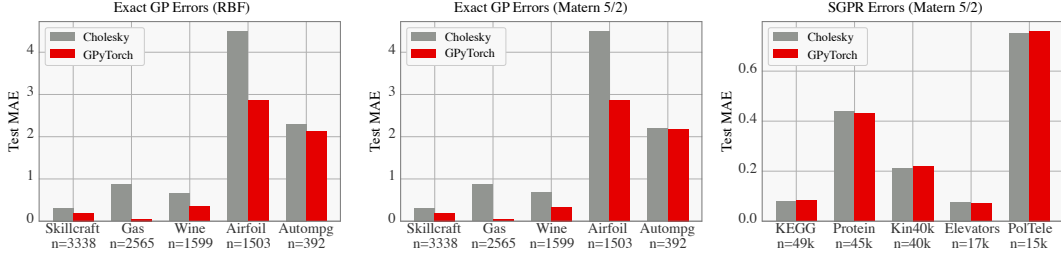

Figure 3: Comparing final Test MAE when using BBMM versus Cholesky based inference. The left two plots compare errors using Exact GPs with RBF and Matern-5/2 kernels, and the final plot compares error using SGPR with a Matern-5/2 kernel on significantly larger datasets.

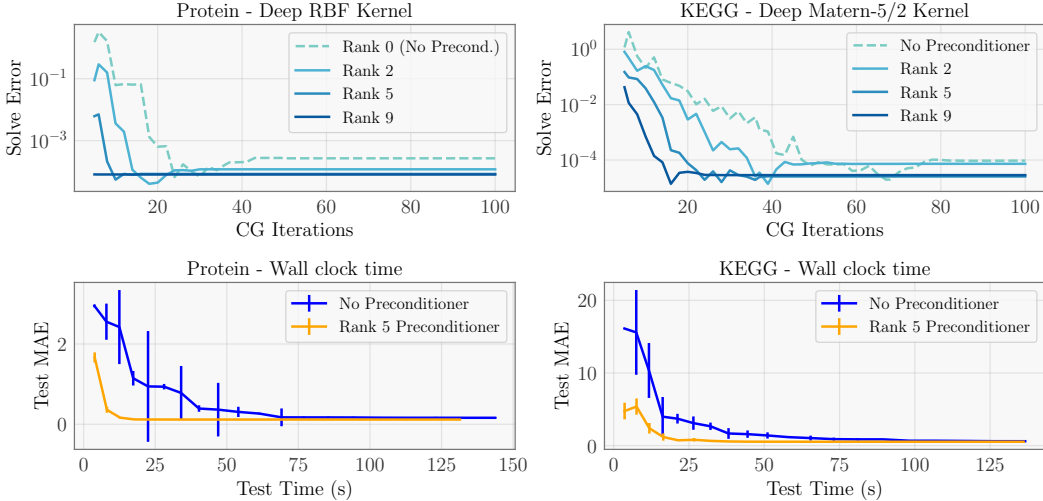

Figure 4: The effect of preconditioning on solve errors $\|K\mathbf{x}^* - \mathbf{y}\|/\|\mathbf{y}\|$ achieved by linear conjugate gradients using no preconditioner versus rank 2, 5, and 9 pivoted Cholesky preconditioners on 2 UCI benchmark datasets using deep RBF and deep Matern kernels. The hyperparameters of $K$ were learned by maximizing the marginal log likelihood on each dataset.

celerated Cholesky methods (Exact, SGPR), which only achieve a roughly $4\times$ speedup. This result underscores the fact that Cholesky methods are not as well suited for GPU acceleration. Additionally, BBMM performs better than the GPU-accelerated version of [13] on SKI. This speedup is because BBMM is able to calculate all inference terms in parallel, while [13] computes the terms in series.

**Error comparison.** In Figure 3 we report test mean average error (MAE) for Exact and SGPR models.[3] We demonstrate results using both the RBF kernel and a Matern-5/2 kernel. Across all datasets, our method is at least as accurate in terms of final test MAE. On a few datasets (e.g. Gas, Airfoil, and Wine with Exact GPs) BBMM even improves final test error. CG has a regularizing effects which may improve methods involving the exact kernel over the Cholesky decomposition, where numerical issues resulting from extremely small eigenvalues of the kernel matrix are ignored. For example, Cholesky methods frequently add noise (or "jitter") to the diagonal of the kernel matrix for numerical stability. It is possible to reduce the numerical instabilities with double precision (see Figure 2); however, this requires an increased amount of computation. BBMM on the other hand avoids adding this noise, without requiring double precision.

**Preconditioning.** To demonstrate the effectiveness of preconditioning at accelerating the convergence of conjugate gradients, we first train a deep RBF kernel model on two datasets, Protein and KEGG, and evaluate the solve error of performing $\widehat{K}_{XX}^{-1}\mathbf{y}$ in terms of the relative residual $\|\widehat{K}_{XX}\mathbf{u} - \mathbf{y}\|/\|\mathbf{y}\|$ as a function of the number of CG iterations performed. We look at this error when using no preconditioner, as well as a rank 2, 5, and 9 preconditioner. To demonstrate that the preconditioner is not restricted to use with an RBF kernel, we evaluate using a deep RBF kernel on Protein and a

deep Matern-5/2 kernel on KEGG. The results are in the top of Figure 4. As expected based on our theoretical intuitions for this preconditioner, increasing the rank of the preconditioner substantially reduces the number of CG iterations required to achieve convergence.

In the bottom of Figure 4, we confirm that these more accurate solves indeed have an effect on the final test MAE. We plot, as a function of the total wallclock time required to compute predictions, the test MAE resulting from using no preconditioner and from using a rank 5 preconditioner. The wallclock time is varied by changing the number of CG iterations used to compute the predictive mean. We observe that, because such a low rank preconditioner is sufficient, using preconditioning results in significantly more accurate solves while having virtually no impact on the running time of each CG iteration. Consequentially, we recommend always using the pivoted Cholesky preconditioner with BBMM since it has virtually no wall-clock overhead and rapidly accelerates convergence.

## 7 Discussion

In this paper, we discuss a novel framework for Gaussian process inference (BBMM) based on blackbox matrix-matrix multiplication routines with kernel matrices. We have implemented this framework and several state-of-the-art GP models in our new publicly available GPyTorch package.

**Non-Gaussian likelihoods.** Although this paper primarily focuses on the regression setting, BBMM is fully compatible with variational techniques such as [22, 53], which are also supported in GPyTorch. These approaches require computing the variational lower bound (or ELBO) rather than the GP marginal log likelihood (2). We leave the exact details of the ELBO derivation to other papers (e.g. [22]). However, we note that a single call to mBCG can be used to compute the KL divergence between two multivariate Gaussians, which is the most computationally intensive term of the ELBO.

**Avoiding the Cholesky decomposition.** A surprising and important take-away of this paper is that it is beneficial to avoid the Cholesky decomposition for GP inference, even in the exact GP setting. The basic algorithm for the Cholesky decomposition (described in Appendix C) involves a divide-and conquer approach that can prove ill-suited for parallel hardware. Additionally, the Cholesky decomposition performs a large amount of computation to get a linear solve when fast approximate methods suffice. Ultimately, the Cholesky decomposition of a full matrix takes $\mathcal{O}(n^3)$ time while CG takes $\mathcal{O}(n^2)$ time. Indeed, as shown in Figure 2, CG may even provide *better* linear solves than the Cholesky decomposition. While we use a pivoted version of this algorithm for preconditioning, we only compute the first five rows of this decomposition. By terminating the algorithm very early, we avoid the computational bottleneck and many of the numerical instabilities.

It is our hope that this work dramatically reduces the complexity of implementing new Gaussian process models, while allowing for inference to be performed as efficiently as possible.

## Acknowledgements

JRG and AGW are supported by NSF IIS-1563887 and by Facebook Research. GP and KQW are supported in part by the III-1618134, III-1526012, IIS-1149882, IIS-1724282, and TRIPODS-1740822 grants from the National Science Foundation. In addition, they are supported by the Bill and Melinda Gates Foundation, the Office of Naval Research, and SAP America Inc.

## Footnotes

[2] mBCG differes from Block CG algorithms [35] in that mBCG returns Lanczos tridiagonalization terms.

[3] SKI models are excluded from Figure 3. This is because the BBMM inference engine and the inference engine of Dong et al. [13] return identical outputs (see Appendix A) even though BBMM is faster.

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
