[Supplementary Material]

# Supplementary Information for: GPyTorch: Blackbox Matrix-Matrix Gaussian Process Inference with GPU Acceleration

**Jacob R. Gardner**, **Geoff Pleiss**,
**David Bindel**, **Kilian Q. Weinberger**, **Andrew Gordon Wilson**
Cornell University
{jrg365,kqw4,andrew}@cornell.edu,
{geoff,bindel}@cs.cornell.edu

## A  Analysis of the modified CG algorithm, mBCG

**Linear conjugate gradients** is a widely popular algorithm from numerical linear algebra [12, 17, 42] for rapidly performing matrix solves $A^{-1}\mathbf{b}$. One view of CG is that it obtains matrix solves by solving a quadratic optimization problem:

$$A^{-1}\mathbf{b} = \arg\min_{\mathbf{u}} f(\mathbf{u}) = \arg\min_{\mathbf{u}} \left( \frac{1}{2}\mathbf{u}^\top A\mathbf{u} - \mathbf{u}^\top \mathbf{b} \right). \tag{S1}$$

CG iteratively finds the solution to (S1).   After $n$ iterations, CG is guaranteed in exact arithmetic to find the exact solution to the linear system.   However, each iteration of conjugate gradients produces an approximate solve $\mathbf{u}_k$ that can be computed with a simple recurrence.   We outline this recurrence in Algorithm 1.[4]

---

**Algorithm 1:** Standard preconditioned conjugate gradients (PCG).

---

**Input**  : mvm_A() – function for matrix-vector multiplication (MVM) with matrix $A$
$\qquad\qquad$ **b** – vector to solve against
$\qquad\qquad$ $P^{-1}$() – function for preconditioner
**Output** : $A^{-1}\mathbf{b}$.

$\mathbf{u}_0 \leftarrow \mathbf{0}$ // Current solution
$\mathbf{r}_0 \leftarrow$ mvm_A($\mathbf{u}_0$) - **b** // Current error
$\mathbf{z}_0 \leftarrow P^{-1}(\mathbf{r}_0)$ // Preconditioned error
$\mathbf{d}_0 \leftarrow \mathbf{z}_0$ // "Search" direction for next solution

**for** $j \leftarrow 0$ **to** $T$ **do**
$\quad$ $\mathbf{v}_j \leftarrow$ mvm_A( $\mathbf{d}_{j-1}$ )
$\quad$ $\alpha_j \leftarrow (\mathbf{r}_{j-1}^\top \mathbf{z}_{j-1})/(\mathbf{d}_{j-1}^\top \mathbf{v}_j)$
$\quad$ $\mathbf{u}_j \leftarrow \mathbf{u}_{j-1} + \alpha_j \mathbf{d}_{j-1}$
$\quad$ $\mathbf{r}_j \leftarrow \mathbf{r}_{j-1} - \alpha_j \mathbf{v}_j$
$\quad$ **if** $\|\mathbf{r}_j\|_2 <$ tolerance **then return** $\mathbf{u}_j$ ;
$\quad$ $\mathbf{z}_j \leftarrow P^{-1}($ $\mathbf{r}_j$ $)$
$\quad$ $\beta_j \leftarrow (\mathbf{z}_j^\top \mathbf{z}_j)/(\mathbf{z}_{j-1}^\top \mathbf{z}_{j-1})$
$\quad$ $\mathbf{d}_j \leftarrow \mathbf{z}_j - \beta_j \mathbf{d}_{j-1}$
**end**

**return** $\mathbf{u}_{j+1}$

---

[*]Equal contribution.
[4] Note that this algorithm assumes a preconditioner. In absence of a preconditioner, one can set $P^{-1} = I$.

Conjugate gradients avoids explicitly computing the matrix $A$, and uses only a matrix vector multiply routine instead. This leads to the following observation about CG's running time:

**Observation 1** (Runtime and space of conjugate gradients). *When performing Conjugate gradients to solve $A^{-1}\mathbf{b}$, the matrix $A$ is only accessed through* matrix-vector multiplications *(MVMs) with $A$. The time complexity of $p$ iterations of CG is therefore $\mathcal{O}(p\,\xi(A))$, where $\xi(A)$ is the cost of one MVM with $A$. The space complexity is $\mathcal{O}(n)$ (assuming $A \in \mathbb{R}^{n \times n}$).*

This is especially advantageous is $A$ is a sparse or structured matrix with a fast MVM algorithm. Additionally, CG has remarkable convergence properties, and in practice returns solves accurate to nearly machine precision in $p \ll n$ iterations. The error of the approximation between the $p$th iterate and the optimal solution $\mathbf{u}^*$, $\|\mathbf{u}_k - \mathbf{u}^*\|$, depends more on the conditioning of the matrix than on the size of the matrix $A$. Formally, the error can be bounded in terms of an *exponential decay* involving the *condition number* $\kappa(A) = \|A\|_2\|A^{-1}\|_2$:

**Observation 2** (Convergence of conjugate gradients [12, 17, 42]). *Let $A$ be a positive definite matrix, and let $\mathbf{u}^*$ be the optimal solution to the linear system $A^{-1}\mathbf{b}$. The error of the $p$th iterate of conjugate gradients $\mathbf{u}_p$ can be bounded as follows:*

$$\|\mathbf{u}^* - \mathbf{u}_p\|_A \leq 2\left(\left(\sqrt{\kappa(A)} - 1\right)/\left(\sqrt{\kappa(A)} + 1\right)\right)^p \|\mathbf{u}^* - \mathbf{u}_0\|_A, \tag{S2}$$

where $\|\cdot\|_A$ is the $A$ norm of a vector – i.e. $\|\mathbf{v}\|_A = (\mathbf{v}^\top A \mathbf{v})^{1/2}$ [12, 17, 42].

**Modified Batched Conjugate Gradients (mBCG),** which we introduce in Section 4, makes two changes to standard CG. In particular, it performs multiple solves $A^{-1}B = [A^{-1}\mathbf{b}_1, \ldots, A^{-1}\mathbf{b}_t]$ simultaneously using **matrix-matrix multiplication** (MMM), and it also returns Lanczos tridiagonalization matrices associated with each of the solves. The Lanczos tridiagonalization matrices are used for estimating the log determinant of $A$. mBCG is outlined in Algorithm 2:

---

**Algorithm 2:** Modified preconditioned conjugate gradients (PCG).

---

**Input** : mmm_A() – function for matrix-matrix multiplication with $A$
$\quad\quad\quad$ $B - n \times t$ matrix to solve against
$\quad\quad\quad$ $\widehat{P}^{-1}$() – func. for preconditioner
**Output** : $A^{-1}B$, $\tilde{T}_1, \ldots, \tilde{T}_t$.

$U_0 \leftarrow \mathbf{0}$ // Current solutions
$R_0 \leftarrow$ mmm_A($U_0$) - $B$ // Current errors
$Z_0 \leftarrow P^{-1}(R_0)$ // Preconditioned errors
$D_0 \leftarrow Z_0$ // "Search" directions for next solutions
$\tilde{T}_1, \ldots \tilde{T}_t \leftarrow 0$ // Tridiag matrices
**for** $j \leftarrow 0$ **to** $t$ **do**
$\quad$ $V_j \leftarrow$ mmm_A( $D_{j-1}$ )
$\quad$ $\vec{\alpha}_j \leftarrow (R_{j-1} \circ Z_{j-1})^\top \mathbf{1}/(D_{j-1} \circ V_j)^\top \mathbf{1}$
$\quad$ $U_j \leftarrow U_{j-1}+$ diag($\vec{\alpha}_j$) $D_{j-1}$
$\quad$ $R_j \leftarrow R_{j-1}-$ diag($\vec{\alpha}_j$) $V_j$
$\quad$ **if** $\forall i$ $\left\|\mathbf{r}_j^{(i)}\right\|_2 <$ tolerance **then** **return** $U_j$ ;
$\quad$ $Z_j \leftarrow \widehat{P}^{-1}( R_j)$
$\quad$ $\vec{\beta}_j \leftarrow (Z_j \circ Z_j)^\top \mathbf{1}/(Z_{j-1} \circ Z_{j-1})^\top \mathbf{1}$
$\quad$ $D_j \leftarrow Z_j-$ diag($\vec{\beta}_j$) $D_{j-1}$
$\quad$ $\forall i$ $[\tilde{T}_i]_{j,j} \leftarrow 1/[\vec{\alpha}_j]_i + [\vec{\beta}_{j-1}]_i/[\vec{\alpha}_{j-1}]_i$
$\quad$ $\forall i$ $[\tilde{T}_i]_{j-1,j}, [\tilde{T}_i]_{j,j-1} \leftarrow \sqrt{[\vec{\beta}_{j-1}]_i/[\vec{\alpha}_j]_i}$
**end**
**return** $U_{j+1}$, $\tilde{T}_1, \ldots \tilde{T}_t \leftarrow 0$

---

Note that blue represents an operation that is converted from a vector operation to a matrix operation. red is an addition to the CG algorithm to compute the tridiagonalization matrices.

In this section, we will derive the correctness of the modified batched CG algorithm. We will show that the matrix operations perform multiple solves in parallel. Additionally, we will show that the tridiagonal matrices $\tilde{T}_1, \ldots, \tilde{T}_t$ correspond to the tridiagonal matrices of the Lanczos algorithm [32].

## A.1 Adaptation to multiple right hand sides.

The majority of the lines in Algorithm 2 are direct adaptations of lines from Algorithm 1 to handle multiple vectors simultaneously. We denote these lines in blue. For example, performing

$$V_j \leftarrow \texttt{mmm\_A} ( P_{j-1} )$$

is equivalent to performing $\mathbf{v}_j \leftarrow \texttt{mvm\_A} ( \mathbf{d}_{j-1} )$ for each column of $P_{j-1}$. Thus we can replace multiple MVM calls with a single MMM call.

In standard CG, there are two scalar coefficient used during each iteration: $\alpha_j$ and $\beta_j$ (see Algorithm 1). In mBCG, each solve $\mathbf{u}_1, \ldots, \mathbf{u}_t$ uses different scalar values. We therefore now have *two coefficient vectors*: $\vec{\alpha}_j \in \mathbb{R}^t$ and $\vec{\beta}_j \in \mathbb{R}^t$, where each of the entries corresponds to a single solve. There are two types of operations involving these coefficients:

1. Updates (e.g. $\vec{\alpha}_j \leftarrow (R_{j-1} \circ Z_{j-1})^\top \mathbf{1}/(D_{j-1} \circ V_j)^\top \mathbf{1}$)
2. Scalaing (e.g. $U_j \leftarrow U_{j-1} + \texttt{diag}(\vec{\alpha}_j) \ D_{j-1}$)

The update rules are batched versions of the update rules in the standard CG algorithm. For example:

$$\begin{bmatrix} [\vec{\alpha}_j]_1 \\ \vdots \\ [\vec{\alpha}_j]_t \end{bmatrix} = \frac{(R_{j-1} \circ Z_{j-1})^\top \mathbf{1}}{(D_{j-1} \circ V_j)^\top \mathbf{1}} = \begin{bmatrix} \frac{([R_{j-1}]_1 \circ [Z_{j-1}]_1)\mathbf{1}}{([D_{j-1}]_1 \circ [V_j]_1)\mathbf{1}} \\ \vdots \\ \frac{([R_{j-1}]_t \circ [Z_{j-1}]_t)\mathbf{1}}{([D_{j-1}]_t \circ [V_j]_t)\mathbf{1}} \end{bmatrix} = \begin{bmatrix} \frac{[R_{j-1}]_1^\top [Z_{j-1}]_1}{[D_{j-1}]_1^\top [V_j]_1} \\ \vdots \\ \frac{[R_{j-1}]_t^\top [Z_{j-1}]_t}{[D_{j-1}]_t^\top [V_j]_t} \end{bmatrix},$$

using the identity $(\mathbf{v} \cdots \mathbf{v}')\mathbf{1} = \mathbf{v}^\top \mathbf{v}'$. Thus these updates are batched versions of their non-batched counterparts in Algorithm 1. Similarly, for scaling,

$$\begin{bmatrix} [U_j]_1 & \cdots & [U_j]_t \end{bmatrix} = U_j = U_{j-1} + \text{diag}(\alpha_j)D_{j-1}$$
$$= \begin{bmatrix} [U_{j-1}]_1 & \cdots & [U_{j-1}]_t \end{bmatrix} + \begin{bmatrix} [\alpha_j]_1 [D_{j-1}]_1 & \cdots & [\alpha_j]_t [D_{j-1}]_t \end{bmatrix}.$$

This these scaling operations are also batched versions of their counterparts in Algorithm 1. mBCG is therefore able to perform all solve operations in batch, allowing it to perform multiple solves at once.

## A.2 Obtaining Lanczos tridiagonal matrices from mBCG.

To motivate the Lanczos tridiagonal matrices $\tilde{T}_1, \ldots, \tilde{T}_t$ from mBCG, we will first discuss the Lanczos algorithm. Then, we will discuss how mBCG recovers these matrices.

**The Lanczos algorithm** [32] is an iterative MVM-based procedure to obtain a *tridiagonalization* of a symmetric matrix $A$. A tridiagonalization is a decomposition $A = QTQ^\top$ with $Q \in \mathbb{R}^{n \times n}$ orthonormal and $T \in \mathbb{R}^{n \times n}$ tridiagonal – i.e.

$$T = \begin{bmatrix} d_1 & s_1 & & & & 0 \\ s_1 & d_2 & s_2 & & & \\ & s_2 & d_3 & \ddots & & \\ & & \ddots & \ddots & s_{n-1} \\ 0 & & & s_{n-1} & d_n \end{bmatrix}. \tag{S3}$$

The exact $Q$ and $T$ matrices are uniquely determined by a *probe vector* $\mathbf{z}$ – which is the first column of $Q$. The Lanczos algorithm iteratively builds the rest of $Q$ and $T$ by forming basis vectors in the *Krylov subspace* – i.e.

$$\text{span} \left\{ \mathbf{z}, A\mathbf{z}, A^2\mathbf{z}, \ldots, A^{n-1}\mathbf{z} \right\}, \tag{S4}$$

and applying Gram-Schmidt orthogonalization to these basis vectors. The orthogonalized vectors are collected in $Q$ and the Gram-Schmidt coefficients are collected in $T$. Lanczos [32] shows that $n$

iterations of this procedure produces an exact tridiagonalization $A = QTQ^\top$. $p$ iterations yields a low-rank *approximate tridiagonalization* $A \approx \tilde{Q}\tilde{T}\tilde{Q}^\top$, where $\tilde{Q} \in \mathbb{R}^{n \times p}$ and $\tilde{T} \in \mathbb{R}^{p \times p}$.

**Connection between the Lanczos algorithm and conjugate gradients.**

There is a well-established connection between the Lanczos algorithm and conjugate gradients [12, 17, 42]. In fact, the conjugate gradients algorithm can even be *derived* as a byproduct of the Lanczos algorithm. Saad [42] and others show that it is possible to recover the $\tilde{T}$ tridiagonal Lanczos matrix by *reusing coefficients* generated in CG iterations. In particular, we will store the $\alpha_j$ and $\beta_j$ coefficients from Algorithm 1.

**Observation 3** (Recovering Lanczos tridiagonl matrices from standard CG [42])**.** *Assume we use $p$ iterations of standard preconditioned conjugate gradients to solve $A^{-1}\mathbf{z}$ with preconditioner $P$. Let $\alpha_1, \ldots, \alpha_p$ and $\beta_1, \ldots, \beta_p$ be the scalar coefficients from each iteration (defined in Algorithm 1). The matrix*

$$
\begin{bmatrix}
\frac{1}{\alpha_1} & \frac{\sqrt{\beta_1}}{\alpha_1} & & & & 0 \\
\frac{\sqrt{\beta_1}}{\alpha_1} & \frac{1}{\alpha_2} + \frac{\beta_1}{\alpha_1} & \frac{\sqrt{\beta_2}}{\alpha_2} & & & \\
& \frac{\sqrt{\beta_2}}{\alpha_2} & \frac{1}{\alpha_3} + \frac{\beta_2}{\alpha_2} & \frac{\sqrt{\beta_3}}{\alpha_3} & & \\
& & \ddots & \ddots & \frac{\sqrt{\beta_{m-1}}}{\alpha_{m-1}} \\
0 & & & \frac{\sqrt{\beta_{m-1}}}{\alpha_{m-1}} & \frac{1}{\alpha_m} + \frac{\beta_{m-1}}{\alpha_{m-1}}
\end{bmatrix}
\tag{S5}
$$

*is equal to the Lanczos tridiagonal matrix $\tilde{T}$, formed by running $p$ iterations of Lanczos to achieve $\tilde{Q}^\top P^{-1}A\tilde{Q} = \tilde{T}$) with probe vector $\mathbf{z}$.*

(See [42], Section 6.7.3.) In other words, we can recover the Lanczos tridiagonal matrix $\tilde{T}$ simply by running CG. Our mBCG algorithm simply exploits this fact. The final two lines in red in Algorithm 2 use the $\vec{\alpha}_j$ and $\vec{\beta}_j$ coefficients to form $t$ tridiagonal matrices. If we are solving the systems $A^{-1}[\mathbf{b}_1, \ldots, \mathbf{b}_t]$, then the resulting tridiagonal matrices correspond to the Lanczos matrices with probe vectors $\mathbf{b}_1, \ldots, \mathbf{b}_t$.

## B  Runtime analysis of computing inference terms with mBCG

We first briefly analyze the running time of mBCG (Algorithm 2) itself. The algorithm performs matrix multiplies with $\widehat{K}_{XX}$ once before the loop and once during every iteration of the loop. Therefore, the running time of mBCG is at least $\mathcal{O}(p\Xi(\widehat{K}_{XX}))$, where $\Xi(\widehat{K}_{XX})$ is the time to multiply $\widehat{K}_{XX}$ by an $n \times t$ matrix.

For the remainder of the algorithm, all matrices involved ($U_j, V_j, R_j, Z_j, P_j$) are $n \times t$ matrices. All of the lines involving only these matrices perform operations that require $\mathcal{O}(nt)$ time. For example, elementwise multiplying $Z_j \circ Z_j$ accesses each element in $Z_j$ once, and and then multiplying it by the vector of ones similarly accesses every element in the matrix once. Multiplying $V_j$ by the diagonal matrix with $\mathbf{a}_j$ on the diagonal takes $\mathcal{O}(nt)$ time, because we multiply every element $[V_j]_{ik}$ by $[\mathbf{a}_j]_i$. Therefore, all other lines in the algorithm are dominated by the matrix multiply with $\widehat{K}_{XX}$, and the total running time is also $\mathcal{O}(p\Xi(\widehat{K}_{XX}))$. Furthermore, because these intermediate matrices are $n \times t$, the space requirement (beyond what is required to store $\widehat{K}_{XX}$) is also $\mathcal{O}(nt)$.

We will now show that, after using mBCG to produce the solves and tridiagonal matrices, recovering the three inference terms takes little additional time and space. To recap, we run mBCG to recover

$$
\begin{bmatrix} \mathbf{u}_0 & \mathbf{u}_1 & \cdots & \mathbf{u}_t \end{bmatrix} = \widehat{K}_{XX}^{-1} \begin{bmatrix} \mathbf{y} & \mathbf{z}_1 & \cdots & \mathbf{z}_t \end{bmatrix} \quad \text{and} \quad \tilde{T}_1, ..., \tilde{T}_t.
$$

where $\mathbf{z}_i$ are random vectors and $\tilde{T}_i$ are their associated Lanczos tridiagonal matrices.

**Time complexity of $\widehat{K}_{XX}^{-1}\mathbf{y}$.** This requires no additional work over running mBCG, because it is the first output of the algorithm.

**Time complexity of $\mathrm{Tr}(\widehat{K}_{XX}^{-1}\frac{d\widehat{K}_{XX}}{d\theta})$.** mBCG gives us access to $\widehat{K}_{XX}^{-1}[\mathbf{z}_1 \ \ldots \ \mathbf{z}_t]$. Recall that we compute this trace as:

$$\mathrm{Tr}\left(\widehat{K}_{XX}^{-1}\frac{dK}{d\theta}\right) \approx \frac{1}{t}\sum_{i=1}^{t}(\mathbf{z}_i\widehat{K}_{XX}^{-1})(\frac{d\widehat{K}_{XX}}{d\theta}\mathbf{z}_i) \tag{S6}$$

We can get $\frac{d\widehat{K}_{XX}}{d\theta}\mathbf{z}_i$ by performing a single matrix multiply $\frac{d\widehat{K}_{XX}}{d\theta}[\mathbf{z}_1 \ldots \mathbf{z}_t]$, requiring $\Xi(\frac{d\widehat{K}_{XX}}{d\theta})$. (We assume that $\Xi(\frac{d\widehat{K}_{XX}}{d\theta}) \approx \Xi(\widehat{K}_{XX})$, which is true for exact GPs and all sparse GP approximations.) After this, we need to perform $t$ inner products between the columns of this result and the columns of $\widehat{K}_{XX}^{-1}[\mathbf{z}_1 \ \ldots \ \mathbf{z}_t]$, requiring $\mathcal{O}(tn)$ additional time. Therefore, the running time is still dominated by the running time of mBCG. The additional space complexity involves the $2t$ length $n$ vectors involved in the inner products, which is negligible.

**Time complexity of $\log|\widehat{K}_{XX}|$.** mBCG gives us $p \times p$ tridiagonal matrices $\tilde{T}_1, ..., \tilde{T}_t$. To compute the log determinant estimate, we must compute $e_1^\top \log \tilde{T}_i e_1$ for each $i$. To do this, we eigendecompose $\tilde{T}_i = V_i\Lambda_i V_i^\top$, which can be done in $\mathcal{O}(p^2)$ time for tridiagonal matrices, and compute

$$e_1^\top V_i \log \Lambda_i V_i^\top e_1 \tag{S7}$$

where now the $\log$ is elementwise over the eigenvalues. Computing $V_i^\top e_1$ simply gets the first row of $V_i$, and $\log \Lambda$ is diagonal, so this requires only $\mathcal{O}(p)$ additional work.

The total running time post-mBCG is therefore dominated by the $\mathcal{O}(tp^2)$ time required to eigendecompose each matrix. This is again significantly lower than the running time complexity of mBCG itself. The space complexity involves storing $2t$ $p \times p$ matrices (the eigenvectors), or $\mathcal{O}(tp^2)$.

## C  The Pivoted Cholesky Decomposition

In this section, we review a full derivation of the *pivoted Cholesky decomposition* as used for precondtioning in our paper. To begin, observe that the standard Cholesky decomposition can be seen as producing a sequentially more accurate low rank approximation to the input matrix $K$. In particular, the Cholesky decomposition algorithm seeks to decompose a matrix $K$ as:

$$\begin{bmatrix} K_{11} & K_{12} \\ K_{12}^\top & K_{22} \end{bmatrix} = \begin{bmatrix} L_{11} & 0 \\ L_{21} & L_{22} \end{bmatrix} \begin{bmatrix} L_{11}^\top & L_{21}^\top \\ 0 & L_{22}^\top \end{bmatrix} \tag{S8}$$

Note that that $K_{11} = L_{11}L_{11}^\top$, $K_{12} = L_{11}L_{21}^\top$, and $K_{22} = L_{21}L_{21}^\top + L_{22}L_{22}^\top$. From these equations, we can obtain $L_{11}$ by recursively Cholesky decomposing $K_{11}$, $L_{21}^\top$ by computing $L_{21}^\top = L_{11}^{-1}K_{12}$, and finally $L_{22}$ by Cholesky decomposing the Schur complement $S = K_{22} - L_{21}L_{21}^\top$.

Rather than compute the full Cholesky decomposition, we can view each iteration of the Cholesky decomposition as producing a slightly higher rank approximation to the matrix $K$. In particular, if $K = \begin{bmatrix} k_{11} & \mathbf{b}^\top \\ \mathbf{b} & K_{22} \end{bmatrix}$, then $L_{11} = \sqrt{k_{11}}$, $L_{21} = \frac{1}{\sqrt{k_{11}}}\mathbf{b}$, and the Schur complement is $S = K_{22} - \frac{1}{k_{11}}\mathbf{b}\mathbf{b}^\top$. Therefore:

$$K = \frac{1}{k_{11}}\begin{bmatrix} k_{11} \\ \mathbf{b} \end{bmatrix}\begin{bmatrix} k_{11} \\ \mathbf{b} \end{bmatrix}^\top + \begin{bmatrix} 0 & 0 \\ 0 & S \end{bmatrix} \tag{S9}$$

$$= \mathbf{q}_1\mathbf{q}_1^\top + \begin{bmatrix} 0 & 0 \\ 0 & S \end{bmatrix}. \tag{S10}$$

Because the Schur complement is positive definite [19], we can continue by recursing on the $n-1 \times n-1$ Schur complement $S$ to get another vector. In particular, if $S = \mathbf{q}_2\mathbf{q}_2^\top + \begin{bmatrix} 0 & 0 \\ 0 & S' \end{bmatrix}$, then:

$$K = \mathbf{q}_1\mathbf{q}_1^\top + \begin{bmatrix} 0 \\ \mathbf{q}_2 \end{bmatrix}\begin{bmatrix} 0 \\ \mathbf{q}_2 \end{bmatrix}^\top + \begin{bmatrix} 0 & 0 \\ 0 & S' \end{bmatrix} \tag{S11}$$

In general, after $k$ iterations, defining $\hat{\mathbf{q}}_i = \begin{bmatrix} \mathbf{0} \\ \mathbf{q}_i \end{bmatrix}$ from this procedure, we obtain

$$K = \sum_{i=1}^{k} \hat{\mathbf{q}}_i \hat{\mathbf{q}}_i^\top + \begin{bmatrix} 0 & 0 \\ 0 & S_k \end{bmatrix}. \tag{S12}$$

The matrix $P_k = \sum_{i=1}^{k} \hat{\mathbf{q}}_i \hat{\mathbf{q}}_i^\top$ can be viewed as a low rank approximation to $K$, with

$$\|K - P_k\|_2 = \left\| \begin{bmatrix} 0 & 0 \\ 0 & S_k \end{bmatrix} \right\|_2.$$

To improve the accuracy of the low rank approximation, one natural goal is to minimize the norm of the Schur complement, $\|S_i\|$, at each iteration. Harbrecht et al. [19] suggest to permute the rows and columns of $S_i$ (with $S_0 = K$) so that the upper-leftmost entry in $S_i$ is the maximum diagonal element. In the first step, this amounts to replacing $K$ with $\pi_1 K \pi_1$, where $\pi_1$ is a permutation matrix that swaps the first row and column with whichever row and column corresponds to the maximum diagonal element of $K$. Thus:

$$\pi_1 K \pi_1 = \mathbf{q}_1 \mathbf{q}_1^\top + \begin{bmatrix} 0 & 0 \\ 0 & S \end{bmatrix}. \tag{S13}$$

To proceed, one can apply the same pivoting rule to $S$ to achieve $\pi_2$. Defining $\hat{\pi}_2 = \begin{bmatrix} 1 & 0 \\ 0 & \pi_2 \end{bmatrix}$, then we have:

$$\hat{\pi}_2 \pi_1 K \pi_1 \hat{\pi}_2 = \hat{\pi}_2 \mathbf{q}_1 \mathbf{q}_1^\top \hat{\pi}_2 + \hat{\mathbf{q}}_2 \hat{\mathbf{q}}_2^\top + \begin{bmatrix} 0 & 0 \\ 0 & S_2 \end{bmatrix}. \tag{S14}$$

To obtain a rank two approximation to $K$ from this, we multiply from the left and right by all permutation matrices involved:

$$K = \pi_1 \mathbf{q}_1 \mathbf{q}_1^\top \pi_1 + \pi_1 \hat{\pi}_2 \hat{\mathbf{q}}_2 \hat{\mathbf{q}}_2^\top \hat{\pi}_2 \pi_1 + E_2. \tag{S15}$$

In general, after $k$ steps, we obtain:

$$K = \sum_{i=1}^{k} (\mathbb{Q}_i \hat{\mathbf{q}}_i)(\mathbb{Q}_i \hat{\mathbf{q}}_i)^\top + E_k, \tag{S16}$$

where $\mathbb{Q}_i = \prod_{j=1}^{i} \hat{\pi}_j$. By collecting these vectors in to a matrix, we have that $K = L_k L_k^\top + E_k$, and thus $K \approx L_k L_k^\top$.

## C.1 Running time of the pivoted Cholesky decomposition.

Let $L_k L_k^\top$ be the rank $k$ pivoted Cholesky decomposition of $K_{XX}$. We now analyze the time complexity of computing the pivoted Cholesky decomposition and using it for preconditioning. We will prove the claims made about the pivoted Cholesky decomposition made in the main text which are restated here:

**Observation 4** (Properties of the Pivoted Cholesky decomposition)**.**

1. *Let $L_k L_k^\top$ be the rank $k$ pivoted Cholesky decomposition of $K_{XX}$. It can be computed in $\mathcal{O}(\rho(K_{XX})k^2)$ time, where $\rho(K_{XX})$ is the time required to retrieve a single row of $K_{XX}$.*

2. *Linear solves with $\widehat{P} = L_k L_k^\top + \sigma^2 I$ can be performed in $\mathcal{O}(nk^2)$ time.*

3. *The log determinant of $\widehat{P}$ can be computed in $\mathcal{O}(nk^2)$ time.*

**Time complexity of computing $L_k L_k^\top$.** In general, computing $L_k$ requires reading the diagonal of $K_{XX}$ and $k$ rows of the matrix. For a standard positive definite matrix, Harbrecht et al. [19] observes that this amounts to a $\mathcal{O}(nk^2)$ running time. Given that the time requirement for an matrix-vector multiplication with a standard matrix $\widehat{K}_{XX}$ is $\mathcal{O}(n^2)$, computating the pivoted Cholesky decomposition is a negligible operation.

More generally if we wish to avoid computing the exact matrix $K_{XX}$, then the time requirement is $\mathcal{O}(\rho(A)k^2)$, where $\rho(A)$ is the time required to access a row of $A$. When applying the SoR approximation ($K_{XX} = K_{XU}K_{UU}^{-1}K_{XU}^{\top}$), we have that $\rho(K) = \mathcal{O}(nm)$. Thus, for SGPR, the time complexity of computing the rank $k$ pivoted Cholesky decomposition is $\mathcal{O}(nmk^2)$ time. Assuming that $k^2 \leq m$ or $k^2 \approx m$, this operation will cost roughly the same as a single MVM.

When applying the SKI approximation ($K_{XX} = WK_{UU}W^{\top}$), we have that $\rho(K_{XX}) = \mathcal{O}(n)$. In particular, the ith row of $K_{XX}$ is given by $[K_{XX}]_i = \mathbf{w}_i K_{UU}W^{\top}$. Rather than explicitly perform these multiplications using Toeplitz matrix arithmetic, we observe that $\mathbf{w}_i K_{UU}$ is equivalent to summing four elements from each column of $K_{UU}$ (corresponding to the four non-zero elements of $\mathbf{w}_i$). Since elements of $K_{UU}$ can be accessed in $\mathcal{O}(1)$ time, this multiplication requires $\mathcal{O}(m)$ work. After computing $\mathbf{v} = \mathbf{w}_i K_{UU}$, computing $\mathbf{v}W^{\top}$ requires $\mathcal{O}(n)$ work due to the sparsity of $W^{\top}$. Therefore, we can compute a pivoted Cholesky decomposition for a SKI kernel matrix in $\mathcal{O}(nk^2)$ time. This time complexity is comparable to the MVM time complexity, which is also linear in $n$.

**Time complexity of computing $\widehat{P}_k^{-1}\mathbf{y}$ and $\log|\widehat{P}_k|$.** To compute solves with the preconditioner, we make use of the Woodbury formula. Observing that $P_k = L_k L_k^{\top}$,

$$\widehat{P}_k^{-1}\mathbf{y} = (L_k L_k^{\top} + \sigma^2 I)^{-1}\mathbf{y} = \frac{1}{\sigma^2}\mathbf{y} - \frac{1}{\sigma^4}L_k(I - \frac{1}{\sigma^2}L_k^{\top}L_k)^{-1}L_k^{\top}\mathbf{y}$$

Computing $L_k^{\top}\mathbf{y}$ takes $\mathcal{O}(nk)$ time. After computing the $k \times k$ matrix $I - \frac{1}{\sigma^2}L_k^{\top}L_k$ in $\mathcal{O}(nk^2)$ time, computing a linear solve with it takes $\mathcal{O}(k^3)$ time. Therefore, each solve with the preconditioner, $P_k + \sigma^2 I$, requires $\mathcal{O}(nk^2)$ time total. To compute the log determinant of the preconditioner, we make use of the matrix determinant lemma:

$$\log|\widehat{P}_k| = \log|P_k + \sigma^2 I| = \log|I - \frac{1}{\sigma^2}L_k^{\top}L_k| + 2n\log\sigma$$

Since $I - \frac{1}{\sigma^2}L_k L_k^{\top}$ is a $k \times k$ matrix, we can compute the above log determinant in $\mathcal{O}(nk^2)$ time.

## D  Convergence Analysis of Pivoted Cholesky Preconditioned CG

In this section we prove Theorem 1, which bounds the convergence of pivoted Cholesky-preconditioned CG for univariate RBF kernels.

**Theorem 1 (Restated).** *Let $K_{XX} \in \mathbb{R}^{n \times n}$ be a $n \times n$ univariate RBF kernel, and let $L_k L_k^{\top}$ be its rank $k$ pivoted Cholesky decomposition. Assume we are using preconditioned CG to solve the system $\widehat{K}_{XX}^{-1}\mathbf{y} = (K_{XX} + \sigma^2 I)^{-1}\mathbf{y}$ with preconditioner $\widehat{P} = (L_k L_k^{\top} + \sigma^2 I)$. Let $\mathbf{u}_k$ be the $k^{th}$ solution of CG, and let $\mathbf{u}^* = \widehat{K}_{XX}^{-1}\mathbf{y}$ be the exact solution. Then there exists some $b > 0$ such that:*

$$\|\mathbf{u}^* - \mathbf{u}_k\|_{\widehat{K}_{XX}} \leq 2\left(\frac{1}{1 + \mathcal{O}(n^{-1/2}\exp(kb/2))}\right)^p \|\mathbf{u}^* - \mathbf{u}_0\|_{\widehat{K}_{XX}}. \tag{S17}$$

*Proof.* Let $L_k L_k^{\top}$ be the rank $k$ pivoted Cholesky decomposition of a univariate RBF kernel $K_{XX}$. We begin by stating a well-known CG convergence result, which bounds error in terms of the *conditioning number* $\kappa$ (see Observation 2):

$$\|\mathbf{u}^* - \mathbf{u}_k\|_{\widehat{K}_{XX}} \leq 2\left(\frac{\sqrt{\kappa\left(\widehat{P}_k^{-1}\widehat{K}_{XX}\right)} - 1}{\sqrt{\kappa\left(\widehat{P}_k^{-1}\widehat{K}_{XX}\right)} + 1}\right)^p \|\mathbf{u}^* - \mathbf{u}_0\|_{\widehat{K}_{XX}}. \tag{S18}$$

Our goal is therefore to bound the condition number of $(L_k L_k^{\top} + \sigma^2 I)^{-1}(K_{XX} + \sigma^2 I)$. To do so, we will first show that $L_k L_k^{\top}$ rapidly converges to $K_{XX}$ as the rank $k$ increases. We begin by restating the primary convergence result of [19]:

**Lemma 2** (Harbrecht et al. [19]). *If the eigenvalues of a positive definite matrix $K_{XX} \in \mathbb{R}^{n \times n}$ satisfy $4^k \lambda_k \lesssim \exp(-bk)$ for some $b > 0$, then the rank $k$ pivoted Cholesky decomposition $L_k L_k^{\top}$ satisfies*

$$Tr(K_{XX} - L_k L_k^{\top}) \lesssim n\exp(-bk).$$

(See Harbrecht et al. [19] for proof.) Intuitively, if the eigenvalues of a matrix decay very quickly (exponentially), then it is very easy to approximate with a low rank matrix, and the pivoted Cholesky algorithm rapidly constructs such a matrix. While there has been an enormous amount of work understanding the eigenvalue distributions of kernel functions (e.g., [48]), in this paper we prove the following useful bound on the eigenvalue distribution of univariate RBF kernel matrices:

**Lemma 3.** *Given $x_1, \ldots, x_n \in [0, 1]$, the univariate RBF kernel matrix $K_{XX} \in \mathbb{R}^{n \times n}$ with $K_{ij} = \exp\left(-\gamma(x_i - x_j)^2\right)$ has eigenvalues bounded by:*

$$\lambda_{2l+1} \leq 2ne^{-\gamma/4} I_{l+1}(\gamma/4) \sim \frac{2ne^{-\gamma/4}}{\sqrt{\pi\gamma}} \left(\frac{e\gamma}{8(l+1)}\right)^{l+1}$$

*where $I_j$ denotes the modified Bessel function of the first kind with parameter $j$.*

(See Appendix E for proof.) Thus, the eigenvalues of an RBF kernel matrix $K_{XX}$ decay *super-exponentially*, and so the bound given by Lemma 2 applies.

Lemma 3 lets us argue for the pivoted Cholesky decomposition as a preconditioner. Intuitively, this theorem states that the pivoted Cholesky $L_k L_k$ converges rapidly to $K_{XX}$. Alternatively, the preconditioner $(L_k L_k^\top + \sigma^2 I)^{-1}$ converges rapidly to $\widehat{K}_{XX}^{-1} = (K_{XX} + \sigma^2 I)^{-1}$ – the optimal preconditioner in terms of the number of CG iterations. We explicitly relate Lemma 3 to the rate of convergence of CG by bounding the condition number:

**Lemma 1 (Restated).** *Let $K_{XX} \in \mathbb{R}^{n \times n}$ be a univariate RBF kernel matrix. Let $L_k L_k^\top$ be the rank $k$ pivoted Cholesky decomposition of $K_{XX}$, and let $\widehat{P}_k = L_k L_k^\top + \sigma^2 I$. Then there exists a constant $b > 0$ so that the condition number $\kappa(\widehat{P}^{-1} \widehat{K}_{XX})$ satisfies the following inequality:*

$$\kappa\left(\widehat{P}_k^{-1} \widehat{K}_{XX}\right) \triangleq \left\|\widehat{P}_k^{-1} \widehat{K}_{XX}\right\|_2 \left\|\widehat{K}_{XX}^{-1} \widehat{P}_k\right\|_2 \leq (1 + \mathcal{O}(n \exp(-bk)))^2 . \tag{S19}$$

(See Appendix E for proof.) Lemma 1 lets us directly speak about the impact of the pivoted Cholesky preconditioner on CG convergence. Plugging Lemma 1 into standard CG convergence bound (S18):

$$\|\mathbf{u}^* - \mathbf{u}_k\|_{\widehat{K}_{XX}} \leq 2 \left(\frac{\sqrt{\kappa\left(\widehat{P}_k^{-1} \widehat{K}_{XX}\right)} - 1}{\sqrt{\kappa\left(\widehat{P}_k^{-1} \widehat{K}_{XX}\right)} + 1}\right)^p \|\mathbf{u}^* - \mathbf{u}_0\|_{\widehat{K}_{XX}}$$

$$\leq 2 \left(\frac{1 + \mathcal{O}(n \exp(-bk)) - 1}{1 + \mathcal{O}(n \exp(-bk)) + 1}\right)^p \|\mathbf{u}^* - \mathbf{u}_0\|_{\widehat{K}_{XX}}$$

$$= 2 \left(\frac{1}{1 + \mathcal{O}(\exp(kb)/n)}\right)^p \|\mathbf{u}^* - \mathbf{u}_0\|_{\widehat{K}_{XX}} .$$

$\square$

# E Proofs of Lemmas

## E.1 Proof of Lemma 1

*Proof.* Let $K_{XX} \in \mathbb{R}^{n \times n}$ be a univariate RBF kernel matrix, and let $L_k L_k^\top$ be its rank $k$ pivoted Cholesky decomposition. Let $E$ be the difference between $K_{XX}$ and its low-rank pivoted Cholesky approximation – i.e. $E = K_{XX} - L_k L_k^\top$. We have:

$$\kappa\left(\widehat{P}_k^{-1} \widehat{K}_{XX}\right) \triangleq \left\|\widehat{P}_k^{-1} \widehat{K}_{XX}\right\|_2 \left\|\widehat{K}_{XX}^{-1} \widehat{P}_k\right\|_2$$

$$= \left\|\left(L_k L_k^\top + \sigma^2 I\right)^{-1} \left(K_{XX} + \sigma^2 I\right)\right\|_2 \left\|\left(L_k L_k^\top + \sigma^2 I\right) \left(K_{XX} + \sigma^2 I\right)^{-1}\right\|_2$$

$$= \left\|\left(L_k L_k^\top + \sigma^2 I\right)^{-1} \left(L_k L_K^\top + E + \sigma^2 I\right)\right\|_2 \left\|\left(K_{XX} - E + \sigma^2 I\right) \left(K_{XX} + \sigma^2 I\right)^{-1}\right\|_2$$

$$= \left\|I + \left(L_k L_k^\top + \sigma^2 I\right)^{-1} E\right\|_2 \left\|I - \left(K_{XX} + \sigma^2 I\right)^{-1} E\right\|_2$$

Applying Cauchy-Schwarz and the triangle inequality we have

$$\kappa\left(\widehat{P}_k^{-1}\widehat{K}_{XX}\right) \leq \left(1 + \left\|\left(L_kL_k^\top + \sigma^2 I\right)^{-1}\right\|_2 \|E\|_2\right)\left(1 + \left\|\left(K_{XX} + \sigma^2 I\right)^{-1}\right\|_2 \|E\|_2\right)$$

Let $c$ be some constant such that $c \geq \left\|\left(L_kL_k^\top + \sigma^2 I\right)^{-1}\right\|_2$ and $c \geq \left\|\left(K_{XX} + \sigma^2 I\right)^{-1}\right\|_2$. Then:

$$\kappa\left(\widehat{P}_k^{-1}\widehat{K}_{XX}\right) \leq (1 + c\|E\|_2)^2$$

Harbrecht et al. [19] show that $E$ is guaranteed to be positive semi-definite, and therefore $\|E\|_2 \leq \text{Tr}(E)$. Recall from Lemma 2 and Lemma 3 that $\text{Tr}(E) = \text{Tr}(K_{XX} - L_kL_k^\top) \lesssim n\exp(-bk)$ for some $b > 0$. Therefore:

$$\kappa\left(\widehat{P}_k^{-1}\widehat{K}_{XX}\right) \leq (1 + \mathcal{O}(n\exp(-bk)))^2.$$

$\square$

### E.2 Proof of Lemma 3

*Proof.* We organize the proof into a series of lemmata. First, we observe that if there is a degree $d$ polynomial that approximates $\exp(-\gamma r^2)$ to within some $\epsilon$ on $[-1, 1]$, then $\lambda_{d+1}(K_{XX}) \leq n\epsilon$ (Lemma 4). Then in Lemma 5, we show that if $p_l$ is a truncated Chebyshev expansions of degree $2l$, then $|p_l(r) - \exp(-\gamma r^2)| < 2e^{-\gamma/4}I_{l+1}(\gamma/4)$; the argument involves a fact about sums of modified Bessel functions which we prove in Lemma 6. Combining these two lemmas yields the theorem. $\square$

**Lemma 4.** *Given nodes $x_1, \ldots, x_n \in [0, 1]$, define the kernel matrix $K \in \mathbb{R}^{n\times n}$ with $k_{ij} = \phi(x_i - x_j)$. Suppose the degree $d$ polynomial $q$ satisfies $|\phi(r) - q(r)| \leq \epsilon$ for $|r| \leq 1$. Then*

$$\lambda_{d+1}(K) \leq n\epsilon.$$

*Proof.* Define $\tilde{K} \in \mathbb{R}^{n\times n}$ with $\tilde{k}_{ij} = q(x_i - x_j)$. Each column is a sampling at the $X$ grid of a $\deg(q)$ polynomial, so $\tilde{K}$ has rank at most $\deg(q)$. The entries of the error matrix $E = K - \tilde{K}$ are bounded in magnitude by $\epsilon$, so $\|E\|_2 \leq n\epsilon$ (e.g. by Gershgorin's circle theorem). Thus, $\lambda_{d+1}(K) \leq \lambda_{d+1}(\tilde{K}) + \|E\|_2 = n\epsilon.$ $\square$

**Lemma 5.** *For $x \in [-1, 1]$,*

$$|\exp(-\gamma x^2) - p_l(x)| \leq 2e^{-\gamma/4}I_{l+1}(\gamma/4).$$

*Proof.* Given that $|(-1)^j T_{2j}(x)| \leq 1$ for any $x \in [-1, 1]$, the tail admits the bound

$$|\exp(-\gamma x^2) - p_l(x)| \leq 2e^{-\gamma/2}\sum_{j=l+1}^\infty I_j(\gamma/2).$$

Another computation (Lemma 6) bounds the sum of the modified Bessel functions to yield

$$|\exp(-\gamma x^2) - p_l(x)| \leq 2e^{-\gamma/4}I_{l+1}(\gamma/4).$$

$\square$

**Lemma 6.**

$$\sum_{j=l+1}^\infty I_j(\eta) \leq \exp(\eta/2)I_{l+1}(\eta/2)$$

*Proof.* Take the power series expansion

$$I_j(\eta) = \sum_{m=0}^\infty \frac{1}{m!(m+j)!}\left(\frac{\eta}{2}\right)^{2m+j}$$

and substitute to obtain

$$\sum_{j=l+1}^\infty I_j(\eta) = \sum_{j=l+1}^\infty\sum_{m=0}^\infty \frac{1}{m!(m+j)!}\left(\frac{\eta}{2}\right)^{2m+j}.$$

All sums involved converge absolutely, and so we may reorder to obtain

$$\sum_{j=l+1}^{\infty} I_j(\eta) = \sum_{m=0}^{\infty} \frac{1}{m!} \left(\frac{\eta}{2}\right)^m \sum_{j=l+1}^{\infty} \frac{1}{(m+j)!} \left(\frac{\eta}{2}\right)^{m+j}.$$

Because it is the tail of a series expansion for the exponential, we can rewrite the inner sum as

$$\sum_{j=l+1}^{\infty} \frac{1}{(m+j)!} \left(\frac{\eta}{2}\right)^{m+j} = \frac{\exp(\xi_m/2)}{(m+l+1)!} \left(\frac{\xi_m}{2}\right)^{m+l+1}$$

for some $\xi_m$ in $[0, \eta]$, and thus

$$\sum_{j=l+1}^{\infty} \frac{1}{(m+j)!} \left(\frac{\eta}{2}\right)^{m+j} \leq \frac{\exp(\eta/2)}{(m+l+1)!} \left(\frac{\eta}{2}\right)^{m+l+1}.$$

Substituting into the previous expression gives

$$\sum_{j=l+1}^{\infty} I_j(\eta) \leq \sum_{m=0}^{\infty} \frac{1}{m!} \left(\frac{\eta}{2}\right)^m \frac{\exp(\eta/2)}{(m+l+1)!} \left(\frac{\eta}{2}\right)^{m+l+1}$$

$$= \exp\left(\frac{\eta}{2}\right) \sum_{m=0}^{\infty} \frac{1}{m!(m+l+1)!} \left(\frac{\eta}{2}\right)^{2m+l+1}$$

$$= \exp(\eta/2) I_{l+1}(\eta/2).$$

$\square$