[Reviews · NeurIPS 2018]

Reviewer 1



This paper introduces an efficient and general approach to GP inference, which is based on Blackbox Matrix-Matrix multiplication and capitalizes on recent advances in ML hardware. The work is precisely explained, with appropriate references for related and relevant work, and is of great impact. The authors plan to release their work as an easy to use and well maintained software, which would facilitate the adoption of Gaussian process models by the ML community. The key contribution of the paper is a framework that computes GP inference via a blackbox routine that performs matrix-matrix multiplications efficiently with GPU acceleration. It reduces the complexity of implementing Gaussian process models, while still allowing inference to be as efficiently as possible with minimal user intervention. The technical contribution is a modified batched version of linear conjugate gradients, and a method for preconditioning it, which they clearly explain in Section 4 (after providing a solid background, both to GP inference in Section 3 and the key numerical methods involved in Section 2). The authors clearly separate their work from the state of the art: (a) "(...) We utilize the same log determinant estimator (as in [11]); however we avoid explicitly using the Lanczos tridiagonalization algorithm which has storage and numerical stability issues" (b) "Preconditioning is widely acknowledged as an effective tool for accelerating the convergence of conjugate gradients. (...) we do not seek a general purpose preconditioner (...) the preconditioner should afford roughly linear time solves and space. (...) should be able to efficiently compute the log determinant of the preconditioner matrix." Due to the fact that the proposed method reduces the bulk of GP inference to matrix-matrix multiplications, it can be easily adapted to complex GP models or structured GP approximations. They evaluate these claims in Section 6, where they show that (a) the proposed engine provides substantial speed benefit over alternatives (particularly when using GPUs) with comparable inference accuracy, and (b) the proposed preconditioning results in an important improvement in the efficiency of the method. All in all, I argue for the acceptance of this work and encourage the authors to revise the manuscript for minor typos (e.g. "convergence bounds for of our preconditioner" in line 82) and improved clarity (e.g. SLQ is undefined in line 124). The authors have additionally provide interesting insights within their author rebuttal, which I encourage to include in the final manuscript.

Reviewer 2



Summary ======================== The authors propose a generic inference algorithm for GP regression that attempts to exploit the advantages of modern GPU based architectures to perform fast matrix multiplications. They proposed a modification of PCG (preconditioned conjugate gradient descent) to solve “Ab = c” that computes simultaneously the solution for multiple r.h.s vectors it computes, at the same time, the tri-diagonalization of the matrix w.r.t. the r.h.s vectors. Based on these outputs, the author propose efficient estimates of the three main quantities needed for computing the marginal log-likelihood: the projection of the output using the inverse of the kernel matrix , the log-determinant of the kernel matrix and the trace of the kernel matrix times its derivative. In the experimental section they show a significant speed-up with respect to state-of-the-art inference methods for scalable GP regression and improvements in terms of accuracy due to the better stability of the proposal. Details ========================= The paper is focused on scalable inference method for GP regression (although the authors claim the method can be easily extended to other scenarios, e.g. classification). The main idea is to take advantage of modern GPUs and its high performance to perform fast matrix multiplications. To do so, they cleverly modify PCG to solve several r.h.s. vectors simultaneously where the matrix is the kernel matrix of the training set. More specifically, they solve for the output variable "y" and a set of random i.i.d vectors sampled from a standard multivariate gaussian. The random vectors are used to approximate the log determinant term of the kernel matrix in the log-marginal likelihood. In addition, the method outputs the tri-diagonalizations of the kernel matrix w.r.t. the random vectors, that are used to compute the trace term in the log-marginal likelihood. In addition, they propose preconditioning to reduce the number of sequential steps. One of the selling points of the algorithm is that it is black-box: the user only have to define a matrix multiplication operator for the kernel and its derivative. However, the paper focused on regression. It is true that it can be easily extended to other problems like classification, but in that case the algorithm needs specific inference algorithms for the problem at hand and it could be argued that it is not that black-box anymore. In the experimental section the authors perform a wide set of experiments where they show how their method outperforms the baselines in the majority of the scenarios in terms of speed, and in some cases, in terms of accuracy. For the improvement in terms of accuracy the authors hypothesis is that it is related to the better stability of their method, although further research is encouraged. Overall, it is a strong well-written paper with nice theoretical and practical contributions. Minor: Should not be K instead of T at the beginning of eq. (6)?

Reviewer 3



Summary ======= The paper presents a software package for approximate Gaussian process regression with Gaussian likelihood. This software uses a batch version of conjugate gradients to leverage parallelization with GPUs. The implementation can also be used to speed up existing approximation methods like SoR and FITC. To reduce the number of conjugate gradients iterations, partial pivoted Cholesky decomposition is used for preconditioning. The authors provide theoretical analysis how the preconditioner improves upon convergence speed for the RBF kernel. Clarity ======= There is nothing to complain. The language is clear and precise. The paper is easy to understand. Book references (as e.g. [36]) should be augmented with page numbers. Originality =========== The ideas presented in the paper are not ground-breakingly new, but are a clever combination and implementation of existing ones. That is absolutely fine and sufficient for acceptance. Novel is the theoretical analysis of the preconditioner. It is not clear to me whether the modification of conjugate gradients is meant to be novel. If so then the submission lacks references to existing work (the seminal paper "block conjugate gradients" by O'Leary and follow-up literature). I did not look into the appendix to see how both algorithms relate to each other. Quality ======= The presentation of the results regarding speed-up appear arbitrary---twenty conjugate gradient steps are not a lot. I made the experience that in bad cases, conjugate gradients may need more than a thousand steps to arrive at a satisfactory residual. In those cases it is hard to beat the LAPACK implementation of the Cholesky decomposition. The quality of this submission could be improved by adding plots how the approximation quality and the residual evolve over time (for fixed hyper-parameters). This way it is easier to evaluate how much speed-up one can gain, for a certain sacrifice in precision. Significance ============ Facilitating the usage of GPUs for GP inference is an important contribution.